# Effects of Gender and Vitamin D on Vascular Reactivity of the Carotid Artery on a Testosterone-Induced PCOS Model

**DOI:** 10.3390/ijms242316577

**Published:** 2023-11-21

**Authors:** Anita Süli, Péter Magyar, Márton Vezér, Bálint Bányai, Mária Szekeres, Miklós Sipos, Máté Mátrai, Judit Réka Hetthéssy, Gabriella Dörnyei, Nándor Ács, Eszter Mária Horváth, György L. Nádasy, Szabolcs Várbíró, Marianna Török

**Affiliations:** 1Department of Obstetrics and Gynecology, Semmelweis University, 1082 Budapest, Hungary; suli.anita@med.semmelweis-univ.hu (A.S.); sipos.miklos.dr@gmail.com (M.S.); acs.nandor@med.semmelweis-univ.hu (N.Á.); varbiroszabolcs@gmail.com (S.V.); 2Medical Imaging Centre, Faculty of Medicine, Semmelweis University, 1082 Budapest, Hungary; drmagyarpeter@gmail.com; 3Department of Physiology, Faculty of Medicine, Semmelweis University, 1094 Budapest, Hungary; banyai.balint@gmail.com (B.B.); szekeres.maria@semmelweis.hu (M.S.); horvath.eszter@med.semmelweis-univ.hu (E.M.H.); nadasy.gyorgy@med.semmelweis-univ.hu (G.L.N.); 4Department of Morphology and Physiology, Faculty of Health Sciences, Semmelweis University, 1088 Budapest, Hungary; dornyei.gabriella@semmelweis.hu; 5Institute of Translational Medicine, Semmelweis University, 1094 Budapest, Hungary; mate@matrai.org; 6Workgroup of Research Management, Doctoral School, Semmelweis University, 1085 Budapest, Hungary; hetthessy.judit.reka@semmelweis.hu; 7Department of Obstetrics and Gynecology, University of Szeged, 6725 Szeged, Hungary

**Keywords:** gender, carotid artery, vitamin D deficiency, polycystic ovary syndrome, rat model, chronic testosterone treatment, cardiovascular disease

## Abstract

The negative cardiovascular effects of polycystic ovary syndrome (PCOS) and vitamin D deficiency (VDD) have been discussed previously; however, the sex differences between PCOS females and males are not yet known. Our aim was to investigate the effect of PCOS and VDD in the carotid artery of male and female Wistar rats. Females were treated with transdermal testosterone (Androgel) for 8 weeks, which caused PCOS. VDD and vitamin D supplementation were accomplished via diet. The carotid arteries’ contraction and relaxation were examined using myography. Receptor density was investigated using immunohistochemistry. In PCOS females, angiotensin receptor density, angiotensin II-induced contraction, androgen receptor optical density, and testosterone-induced relaxation increased. The increased contractile response may increase cardiovascular vulnerability in women with PCOS. As an effect of VDD, estrogen receptor density increased in all our groups, which probably compensated for the reduced relaxation caused by VDD. Testosterone-induced relaxation was decreased as a result of VDD in males and non-PCOS females, whereas this reduction was absent in PCOS females. Male sex is associated with increased contraction ability compared with non-PCOS and PCOS females. VDD and Androgel treatment show significant gender differences in their effects on carotid artery reactivity. Both VDD and PCOS result in a dysfunctional vascular response, which can contribute to cardiovascular diseases.

## 1. Introduction

Polycystic ovary syndrome (PCOS), also known as Stein Leventhal syndrome, is one of the most common complex endocrine disorders in women of reproductive age; it affects approximately 2–20% of women between 18–44 years. PCOS is often associated with insulin resistance and compensatory hyperinsulinism, obesity (abdominal type), type 2 diabetes mellitus (DM 2), dyslipidemia, and metabolic disorders, therefore significantly increasing the risk of cardiovascular disease (CVD).

The vasorelaxant effect of estrogen may play a role in the reduced prevalence of hypertension in women of reproductive age, and the impairment of this mechanism may contribute to the higher risk of hypertension in women with PCOS [1,2].

Vitamin D deficiency is a serious global public health problem because it increases the risk of many diseases and pathological conditions. It can be associated with many acute and chronic extraosseous diseases, including dental caries, periodontitis, muscle weakness, autoimmune diseases, infectious diseases, cancers (colon, breast, and prostate cancer), metabolic syndrome (obesity and type 2 diabetes mellitus), and several neurological disorders [3]. Vitamin D receptor (VDR) is expressed in most cardiovascular (CV) cell types: vascular smooth muscle cells, endothelial cells, cardiomyocytes, platelets, macrophages, dendritic cells, and some immune cells [4]. Vitamin D deficiency is associated with many cardiovascular diseases, such as hypertension (HT), endothelial dysfunction, coronary artery disease, heart failure, atrial fibrillation, peripheral arterial disease, atherosclerosis, abnormal blood coagulation, myocardial infarction, cardiac hypertrophy, cardiomyopathy, cardiac fibrosis, stroke, and it significantly increases the morbidity and mortality of CVD [5]. 

Several mechanisms have been identified in the pathogenesis of HT, including genetics, signalization pathways modulating vascular smooth muscle function, and the renin–angiotensin–aldosterone system (RAAS) among them [6]. Excessive activity of the latter increases the risk of HT, which can be reduced by blocking the cascade. Close to one billion people worldwide suffer from HT, which is statistically similar to the global frequency of the vitamin D-deficient population. HT occurs more often in winter, among dark-skinned people and people who live far from the equator. It is noteworthy that there is a similar trend regarding the occurrence of vitamin D deficiency [7]. Some researchers have also observed that the expression of renin and the production of angiotensin II (Ang II) is significantly increased in VDR knockout mice [8]. Vitamin D supplementation effectively suppresses renin synthesis, thereby reducing RAAS activity and blood pressure, as well as their complications, including several CVDs [9]. Furthermore, vitamin D deficiency significantly increases the risk of endothelial dysfunction [10]. In addition, under pathogenic conditions, the production of reactive oxygen radicals (ROSs) increases, which leads to increased oxidative stress, which in turn reduces the bioavailability of nitric oxide (NO) through the inhibition of its synthesis and increased degradation [11]. Vitamin D inhibits the activity of the NADPH oxidase enzyme, an important source of ROS production, and in addition, it improves the antioxidant capacity of the endothelium by stimulating the activity of some antioxidant enzymes (superoxide dismutase). Furthermore, vitamin D stimulates NO synthesis in endothelial cells by mediating the activity of endothelial NO synthase (eNOS) [10]. Vitamin D also reduces the risk of endothelial dysfunction by inhibiting other proinflammatory factors such as TNF-alpha and IL-6, which reduce the availability of NO and the activity of eNOS [12]. Endothelial dysfunction is closely related to the etiology and pathogenesis of various CVDs, including diabetic angiopathy, HT, atherosclerosis, and peripheral arterial disease.

Less is known about the potential interplay between vitamin D deficiency and the effects of sexual steroids in vascular pathologies. To study this, we used the previously developed dihydrotestosterone-derived animal model of PCOS. In this model, we investigated PCOS-dependent functional vascular changes in the carotid artery by testing the contraction- and relaxation-dependent effects of certain agonists using the wire myograph technique. 

PCOS and VDD are associated with endothelial dysfunction; the endothelium loses the ability to maintain vascular balance [13]. Endothelial dysfunction can be characterized by reduced bioavailability of nitric oxide, atherosclerosis, and increased oxidative stress [10]. 

In PCOS, insulin resistance, hyperandrogenism, and obesity are potential mediators of endothelial dysfunction [13]. According to a meta-analysis, decreased serum or plasma nitrate levels in PCOS can be measured [14]. Several factors underlie endothelial dysfunction caused by vitamin D deficiency. Through modulating the activity of endothelial NO synthase (eNOS), vitamin D controls the generation of NO in endothelial cells. Reactive oxygen species (ROS) overproduction under pathogenic conditions causes oxidative stress, which promotes NO breakdown and inhibits NO synthesis, lowering NO bioavailability. However, vitamin D increases antioxidant capacity by boosting the activity of antioxidative enzymes like superoxide dismutase and inhibits the production of reactive oxygen species (ROS) via nicotinamide adenine dinucleotide phosphate (NADPH) oxidase. Proinflammatory mediators including TNF-α and IL-6, in addition to ROS, are known to increase the risk of endothelial dysfunction by inhibiting the bioactivity of NO and eNOS and by upregulating the expression of many atherosclerotic factors via the NF-κB pathway. By inhibiting NF-κB signaling and proinflammatory cytokine generation, vitamin D inhibits these proinflammatory activities [10]. Impaired endothelial function may be one of the main causes of the cardiovascular risk-enhancing effects of vitamin D deficiency/PCOS. Our aim was to investigate how the role of certain vascular mediators is altered in PCOS and whether vitamin D treatment can improve PCOS-related functional vascular changes in carotid arteries. In addition, we aimed to study the carotid artery function, and whether sex differences can be detected with respect to vasoconstriction and vasodilatation responses after vitamin D supplementation. To the best of our knowledge, this is the first study in the literature that analyzes the associations between vitamin D and gender and PCOS in the carotid artery.

## 2. Results

### 2.1. Effects of Vitamin D Deficiency and PCOS in Female Groups

Chronic testosterone treatment (transdermal Androgel) increased the Ang II contraction in female animals (FD+T+, FD−T+ groups) (Figure 1). 

Androgel treatment increased the optical density of Angiotensin II type I receptor (ATR1) but only in vitamin D-supplemented female animals (FD^+^T^+^ groups) (Figure 2).

The estrogen-induced relaxation was diminished as an effect of vitamin D deficiency in female groups regardless of Androgel treatment (FD−T− and FD−T+ groups). Indeed, it turned into contraction in the FD−T− group at high concentrations of the agonist (Figure 3).

As a result of vitamin D deficiency, the optical density of the estrogen receptor alpha was considerably increased in female groups (FD−T+, FD−T− groups) (Figure 4).

When female rats were not receiving Androgel treatment, vitamin D deficiency significantly reduced testosterone-induced relaxation (in the FD−T− group), while in FD−T+ animals, vitamin D deficiency did not reduce testosterone-induced relaxation (Figure 5).

The Androgel treatment increased the androgen receptor intensity in female animals. In FD−T+ animals, the androgen receptor optical density was larger compared with the FD−T− animals (Figure 6).

The vitamin D receptor optical density did not differ among either of the female groups studied (VDR OD (arbitrary units): 0.21 ± 0.009 in FD+T− animals, 0.22 ± 0.007 in FD−T− animals, 0.26 ± 0.018 FD+T+ animals and 0.23 ± 0.004 in FD−T+ animals) (not shown on separate Figure). 

### 2.2. Effects of Vitamin D Deficiency in Male Animals

Neither angiotensin II-induced contraction (Figure 1) nor angiotensin receptor intensity (Figure 2) differed between vitamin D-deficient and vitamin D-supplemented males. As a result of vitamin D deficiency, both estrogen-induced and testosterone-induced relaxations were reduced (Figure 3 and Figure 5). However, the estrogen receptor alpha density (Figure 4) and the VDR optical density did not differ among deficient and supplemented male groups (VDR OD, arbitrary unit: 0.20 ± 0.002 in the MD+ group and 0.24 ± 0.01 in the MD− group). 

### 2.3. Sex Differences

Angiotensin II-induced contraction is significantly lower in all female groups (FD+T−, FD−T−, FD+T+, and FD−T+ groups) compared with male (MD− and MD+) rats (Figure 1). Interestingly, the angiotensin receptor density did not show sex differences (Figure 2). In the case of estrogen-induced relaxation, only in the FD−T− group was the relaxation smaller compared with the MD+ animals. Both male and female groups had less estrogen-induced relaxation when vitamin D deficiency was present (Figure 3). Female sex is associated with more pronounced estrogen receptor density regardless of Androgel treatment or vitamin D deficiency (Figure 4). Interestingly, vitamin D deficiency increased the estrogen receptor optical density in male and female animals as well (Figure 4). In the case of testosterone-induced relaxation, only in the FD−T− group was the relaxation smaller compared with the MD+ animals. As a result of vitamin D deficiency, testosterone-induced relaxation turned into contraction in male animals and in females not receiving Androgel treatment (Figure 5). As a result of Androgel treatment or vitamin D supplementation, the relaxation in females was similar to that of vitamin D-supplemented males. The androgen receptor density was significantly higher in the FD−T+ group compared with the FD−T− group (Figure 6). The vitamin D receptor optical density did not differ among the female and male groups.

## 3. Discussion

Both vitamin D deficiency and hyperandrogenic status are known to increase the risk of cardiovascular events, including stroke [15,16,17,18,19,20]. The incidence and severity of strokes show gender differences. Compared with men, women have a higher incidence of stroke, which accumulates at an older age because of women’s longer life expectancies. Men typically have more severe strokes, which result in more fatalities [21,22,23]. Carotid arteries play an important role in the cardiovascular risk. PCOS is also often associated with vitamin D deficiency [24], so both the hyperandrogenic state and vitamin D deficiency can have adverse cardiovascular effects in PCOS women. Previous literature has investigated how vascular function is impaired in animals with PCOS compared with healthy females [25,26]. However, whether the vascular function of PCOS female animals differs from that of healthy males has been less investigated. Vitamin D deficiency impairs vascular function in both healthy and PCOS female animals [27]. At adequate vitamin D levels, vascular function improves in PCOS animals, and we therefore investigated the effects of vitamin D deficiency and supplementation in both PCOS females and males. We investigated how vitamin D deficiency and hyperandrogenism can modify the vasoconstrictor response of the carotid artery using the wire myograph method and what kind of sex differences are found in this process. The originality of the research concept is that we analyzed the effects of testosterone and vitamin D according to gender. We used a hyperandrogenic female PCOS model as described before [25,28]. 

### 3.1. Effects of PCOS on Carotid Artery 

Angiotensin II is an important vasoconstrictor involved in physiological processes in most vessels, including the carotid artery. In our present study, in vitamin-D-supplemented female groups, the angiotensin II receptor optical density was significantly higher in Androgel-treated female animals than in rats not treated with Androgel. In the literature, it has been shown that vasoconstrictor Ang II type 1b receptor expression is increased in the uterine artery following long-term testosterone exposition, whereas vasodilator Ang II type 2 receptor expression is decreased [29]. Increased angiotensin receptor density in our vitamin D-supplemented PCOS animals resulted in an increased vasoconstriction response in our myographic measurements. Interestingly, contractility was also increased in our vitamin D-deficient PCOS female rats, which may be due to androgens enhancing the vascular response to angiotensin II [30,31]; the Ang II-induced vasoconstriction in endothelium-intact and endothelium-denuded uterine arteries of pregnant Sprague–Dawley rats was enhanced by elevated testosterone levels [29]. Furthermore, testosterone treatment increases plasma renin activity in spontaneously hypertensive female rats; renal renin mRNA levels are androgen-dependent in spontaneously hypertensive females [32]. Thus, one possible explanation for the increased cardiovascular risk present in PCOS [33,34] could be that the excessive androgen effect increases angiotensin II response.

Estrogen is an important regulatory hormone in vascular endothelial and smooth muscle cells; estrogen affects vascular tone, antioxidant capacity, anti-inflammatory activity, and reduces blood pressure and atherosclerosis. In our present study, the estrogen receptor alpha density did not change as an effect of chronic Androgel treatments. Changes in the estrogen receptor have been controversial in the literature in response to PCOS. It has been previously described that estrogen receptor expression increases significantly in the stroma and glandular epithelium of women with PCOS. In addition, during the proliferative phase of the menstrual cycle, women with PCOS are reported to have increased levels of endometrial ERα immunostaining in their stroma and epithelium [35]. Even to the contrary, it has been reported that proliferative endometrium in obese PCOS women shows lower mRNA levels of ERα and ERα36, as well as a lower ERα/ERβ mRNA ratio compared with body mass index-matched controls [36,37]. There has been less research in the literature on how estrogen receptor levels change under the influence of PCOS outside the ovaries and endometrium. Estrogen receptor optical density increased in aortic endothelium after chronic Androgel treatment [25]. In our myographic measurements, we did not see any change in estrogen-induced relaxation in our PCOS animals, which could be explained by the fact that we did not see any difference in estrogen receptor density.

The impact of long-term testosterone therapy on androgen receptor density and testosterone-induced relaxation is a fascinating topic. Testosterone acutely leads to vasorelaxation in blood vessels similar to estrogen. In our present study, as an effect of chronic Androgel treatment, androgen receptor OD significantly increased in female animals. Based on literature data, chronic androgen treatment causes upregulation of androgen receptor (AR) gene expression in healthy and PCOS women as well [38,39,40]. In addition, the density of AR in the vagina also depended on the phase of the cycle, with the density of AR being significantly higher in the luteal phase (when serum testosterone is higher) than in the follicular phase (when serum testosterone is lower) [41]. Accordingly, due to this higher AR optical density, in our study the testosterone-induced relaxation was significantly higher in the chronic androgen-treated female animals (FD+T+, FD−T+) than in the FD−T− rats. The increased testosterone-induced relaxation measured in our present study may be due to the increased amount of androgen receptors in the carotid artery.

In summary, in our PCOS female animals, the angiotensin receptor density and angiotensin II-induced contraction increased, the estrogen receptor density and relaxation did not change and the androgen receptor optical density and testosterone-induced relaxation increased as an effect of chronic Androgel treatment. Increased contractile response may increase cardiovascular vulnerability in women with PCOS.

### 3.2. Effects of Vitamin D Deficiency on Carotid Artery 

PCOS is often associated with vitamin D deficiency [24]. As both vitamin D deficiency and PCOS impair vascular function, their co-occurrence further worsens the cardiovascular risk of women with PCOS. Furthermore, vitamin D supplementation significantly improves vascular function in animal models of PCOS [25,42].

Renin–angiotensin–aldosterone system down-regulation by the kidney is mostly dependent on vitamin D. A reduced renin–angiotensin–aldosterone system and anomalies in vascular relaxation brought on by low vitamin D levels may ultimately result in the development of hypertension [43]. In our present study, vitamin D deficiency did not alter angiotensin receptor density in carotid segments in either sex and nor did we see any difference in vasoconstrictor response to vitamin D supplementation or deficiency in our myographic measurements. Vitamin D deficiency affects the Ang II-induced contraction, but the data in the literature are contradictory; previous studies have written about the effect of increasing and decreasing the constrictor response [44,45], depending on the vascular area and the species. It is known from the literature that vitamin D deficiency can lead to high blood pressure and affects the renin–angiotensin system. In vascular smooth muscle cells isolated from vitamin D receptor knockout mice, Angiotensin and Angiotensin II receptor 1 mRNA are significantly upregulated in vitamin D deficiency compared with wild-type mice [8]. Low plasma 25(OH)D concentrations have been linked to an increased risk of hypertension and its related consequences according to a number of observational studies [9,46,47]. Consequently, taking vitamin D supplements appears to be a promising treatment choice for these individuals [9].

By preserving calcium homeostasis, vitamin D controls the synthesis of estrogen in the gonads. In addition, to control vascular processes, estrogen binds to vitamin D receptors on smooth muscle and endothelial cells. Vitamin D stimulates endothelial cells’ and vascular smooth muscle cells’ vitamin D receptors to control endogenous vasodilators such as prostaglandins, nitric oxide, and calcitonin gene-related peptides. NO-mediated endothelium dysfunction is brought on by a vitamin D shortage, and this condition is reduced by vitamin D treatment [43]. Vitamin D and estrogen interact: low vitamin D levels increase the risk of hypoestrogenism [48]. In the present study, chronic vitamin D deficiency resulted in significant changes in estrogen receptor alpha levels: vitamin D deficiency increased estrogen receptor density in PCOS and non-PCOS female groups, and in male animals as well. In the literature, it is contradictory how vitamin D supplementation or deficiency affects estrogen receptor alpha expression. In thoracic aorta cells and coronary artery cells, estrogen receptor expression was decreased by vitamin D deficiency [25,49]. In another study, calcitriol did not significantly affect estrogen receptor α (ERα, ESR1) expression levels in the T47D cells of 3D cultured cells [50]. In contrast, calcitriol down-regulates the expression of ERα and thereby attenuates estrogen signaling in breast cancer cells including the proliferative stimulus provided by estrogens [51]. Surprisingly, despite the increased estrogen receptor density in our study, vitamin D deficiency reduced estrogen-induced relaxation in both PCOS and non-PCOS female and male animals. It is possible that increased receptor density may be a compensatory mechanism for reduced relaxation. It is known from the literature that vitamin D deficiency is known to reduce estrogen-induced relaxation in both sexes [25,49,52]. It is possible that vitamin D deficiency first reduced estrogen-induced relaxation, followed by an increase in carotid artery estrogen receptor-α alpha expression. Chronic elevated testosterone levels affect the estrogen-induced relaxation in different arteries, but the literature is contradictory. Estrogen was found to decrease cerebrovascular tone and increase cerebral blood flow by enhancing endothelial-derived nitric oxide and prostacyclin pathways. Testosterone has the opposite effect, increasing the tone of cerebral arteries [53]. In thoracic aorta segments, the chronic testosterone treatment did not change the estrogen-induced relaxation in female rats [25]. In contrast, the estradiol-dependent vasorelaxation in the dihydrotestosterone-treated rat aorta was significantly reduced compared with the control group [52]. The contradictory results may be due to different types of animals and blood vessels. 

No difference in androgen receptor density was found in our study of the effect of vitamin D deficiency. Few studies in the literature have addressed the changes in endothelial androgen receptors in PCOS and the effect of vitamin D deficiency. It is known that vitamin D treatment in PCOS reduces total testosterone, free testosterone index levels, and also hirsutism. Vitamin D reduces testosterone levels by increasing sexual hormone-binding globulin levels, which can bind extra free testosterone. [54]. In the present study, we found that under the effect of vitamin D deficiency with unchanged androgen receptor density, the testosterone-induced relaxation was decreased in females not receiving Androgel treatment, in response to vitamin D deficiency. There was an interaction between chronic testosterone treatment and vitamin D deficiency. In female animals without testosterone treatment, vitamin D deficiency decreased the testosterone-induced relaxation, but if female rats were treated with testosterone, vitamin D deficiency did not change the testosterone-induced relaxation. This may be due to increased androgen receptor density caused by chronic androgen treatment in FD+T+ groups. In our present study, in males, the testosterone-induced relaxation was smaller in the vitamin D-deficient male group. It was known from previous animal studies, that a vitamin D-deficient state caused diminished testosterone vasodilator capacity of the intramural coronary artery [49], while the androgen receptor optical density remained unchanged [49]. Furthermore, a vitamin D-deficient state caused enhanced testosterone-induced tone (opposite calculation to relaxation) in the cerebral arteries of male rats [55]. Despite the fact that vitamin D deficiency/supplementation had no significant effect on testosterone-induced relaxation in our PCOS rats, it is known from the literature that vitamin D supplementation can help restore vascular reactivity damaged by PCOS [52,55,56,57,58,59]. Endothelial functions improved with vitamin D supplementation in the hyperandrogenic state [56,57,59].

In conclusion, vitamin D deficiency did not affect angiotensin II-induced contraction or angiotensin receptors. Estrogen receptor density increased under vitamin D deficiency in all our groups, which probably compensated for the reduced relaxation caused by vitamin D deficiency. Although vitamin D deficiency did not affect androgen receptor density, it did reduce relaxation in males and non-PCOS females, whereas this reduction was absent in PCOS females. In PCOS females, this reduction in relaxation was lost because of increased AR density and relaxation due to chronic Androgel treatment.

### 3.3. Sex Differences in Carotid Artery

Sex differences in the function of the carotid artery have been previously investigated and may explain the cardiovascular sex difference between men and women. Male sex and vitamin D deficiency both led to an increase in phenylephrine-induced contraction. In male rats, acethylcholine-induced relaxation was smaller compared with females, regardless of vitamin D status. In response to the inhibition of prostanoid signaling, female animals’ ability to contract was decreased, whereas in male rats the relaxation was increased. Furthermore, vitamin D deficiency reduced the elastic fiber density in female rats, but not in males. In contrast, in male animals, vitamin D deficiency reduced the smooth muscle actin and endothelial nitric oxide synthase levels but increased the thromboxane receptor density in the carotid artery. Male sex is associated with decreased nitrative stress compared with females [60].

In the present study, we found sex differences between male and female rat carotid arteries. The significance of the manuscript is enhanced by the fact that we compared not only healthy females but also PCOS females with males.

Although no sex difference in angiotensin receptor optical density was found, angiotensin II-induced contraction was significantly greater in males compared with non-PCOS females (regardless of vitamin D status). There are interactions between sex hormones and components of the renin–angiotensin system for cardiovascular regulation. Estrogen shifts the balance of the renin–angiotensin system toward angiotensin-(1-7)-angiotensin-converting enzyme (ACE)-2 mas receptor-Angiotensin II type 2 receptor, which elicits cardiovascular protection. In contrast in males, testosterone shifts the balance of the renin–angiotensin system toward angiotensin II-ACE-Angiotensin II type 1 receptor pathways, which elicit deleterious cardiovascular actions [61]. In addition, ACE is upregulated by testosterone: older, healthy men have lower ACE activity than younger, healthy men [62]. In middle cerebral arteries, the Ang II-induced contraction is greater in male mice than in female animals due to modulation of Nox2-dependent reactive oxygen species generation. The production of O_2_− and H_2_O_2_ in response to Ang II is higher in cerebral arteries from male wild-type mice compared with females [63]. Furthermore, in basilar cerebral arteries, angiotensin II is a potent vasoconstrictor in male mice but not in females [64]. This gender difference has already been described not only in cerebral but also in renal vessels [45]. 

It is important to point out that, even though Ang II contraction increased following chronic testosterone treatment, Ang II contraction was significantly higher in males compared with PCOS female groups. Ang II-induced vasoconstriction shows gender difference: it is greater in males than in females. It was previously known that Ang II contraction shows a gender difference, but to the best of our knowledge, this is the first study where gender differences were investigated between females of the PCOS model and males. In our PCOS model, the vascular reactivity of female animals was midway between vitamin D-treated males and control females during angiotensin II-induced contraction. These differences observed in Ang II-induced contraction can be also explained by the previously mentioned fact that androgens (males and PCOS females in our present study) enhance and estrogens decrease the vascular response to angiotensin II [30,31,45]. 

In the present study, chronic vitamin D deficiency resulted in significant changes in estrogen receptor alpha levels, and significant sex differences in these processes were also observed. As a sex difference, it should be emphasized that estrogen receptor density was lower in males compared with non-PCOS and PCOS vitamin D-deficient females. The sex differences between estrogen-relaxation and receptor expression are well known in the literature [65,66]. Interestingly, in our present study as a recorded sex difference, VDD females not receiving Androgel treatment differed from vitamin D-supplemented males. 

Although we did not see a sex difference in androgen receptor density, testosterone-induced relaxation under chronic androgen treatment became similar to that of vitamin D-supplemented males. In our present study, the testosterone-induced relaxation was most intensive in the vitamin D-supplemented male group, the testosterone-induced relaxation was significantly higher in MD+ groups compared with the vitamin D-deficient group and androgen-untreated female group (FD−T−). There are studies in the literature that discuss the different relaxant effects of testosterone in male and female animals. Testosterone-induced relaxation was greater in males than in females with intact endothelium (normal). In deendothelized coronaries, testosterone had no relaxing effect in males, whereas in females there was a measurable relaxation. This suggests that testosterone-induced relaxation is normally lower in females than in males (with intact endothelium). In addition, in males mainly endothelium-dependent NO-mediated relaxation is involved, while in females other non-genomic pathways are also involved [67]. 

In summary, angiotensin receptor density showed no difference, while angiotensin II-induced contraction was significantly greater in males compared with non-PCOS and PCOS female rats, possibly due to the activation of an enzyme of the renin–angiotensin system. Estrogen receptor density was significantly lower in males compared with vitamin D-deficient females, while with regard to sex differences, the estrogen-induced relaxation differed in only VDD females not receiving Androgel treatment from vitamin-D-supplemented males. In VDD and non-PCOS females, the testosterone-induced relaxation was smaller compared with the vitamin D-supplemented males. As a result of Androgel treatment or vitamin D supplementation, the relaxation in females is similar to that of vitamin D-supplemented males. 

### 3.4. Strengths and Limitations

This is the first comparison of female, hyper-androgenic female, and male vascular reactivity on the effect of testosterone in females and males. In some respects, hyperandrogenic females showed intermediate reactivity between intact females and males; however, some individual reactions were also described here. We note that the results of animal studies should be interpolated to humans with limitations. Because of some methodological problems, the immunohistochemical analysis had lower case numbers. However, our results might inspire human studies in this field.

## 4. Materials and Methods

### 4.1. Chemicals

The experimental polycystic ovary syndrome was achieved by transdermal testosterone treatment (Androgel 30 × 50 mg, Besins Manufacturing Belgium, Groot Bijgaardenstraat 128, 1620 Drogenbos, Belgium) [28].

Treatment with cholecalciferol was performed with Vigantol (Manufacturer: Merck/Merck Serono Business Unit, Merck Ltd., Budapest) 20,000 IU/mL drops [28]. 

During surgical procedures, pentobarbital (Nembutal, ASTfarma 50 mg/kg i.p.) was used for anesthesia. 

Krebs solution was used for in vitro studies. The composition of the Krebs solution is as follows: (amounts are in mM/L) (119 NaCl, 4.7 KCl, 2.5 CaCl_2_·2H_2_O, 1.17 MgSO_4_·7H_2_O, 24 NaHCO_3_, 1.18 KH_2_PO_4_, 1.2 NaH_2_PO_4_, 0.034) EDTA, and 5,5 glucose (Sigma-Aldrich Co, St. Louis, MO, USA and Budapest, Hungary). The solution was warmed to 37 °C, pH maintained at 7.4, and it was perfused with a carbogen gas containing 5% carbon dioxide and 95% oxygen. Phenylephrine (Phe), Angiotensin II (Ang II), DMSO, testosterone, and estrogen were purchased from Sigma-Aldrich (St. Louis, MO, USA, and Budapest, Hungary). Chemicals were freshly dissolved in physiological saline (0.9% *v*/*v* NaCl) or Krebs solution on the day of the experiment.

### 4.2. Animals

There were no medical or surgical complications. All procedures complied with legal and institutional requirements for animal health (Guide for the Care and Use of Laboratory Animals, NIH -1996, the US National Institutes of Health, 8th edition, 2011, and the EU-conformed Hungarian Law on Animal Care, XXVIII/1998). The institutional Animal Care Commission and National authorities accepted the research protocol (IRB: PEI/001/820-2/2015).

For our experiments, 4-week-old (21–28 days old) female and male Wistar rats, weighing 100–140 gr (Charles River Ltd., Animalab, Vác, Hungary–Animal Facility of Semmelweis University, Budapest) were randomly divided into six experimental groups, namely, female vitamin D-supplemented group, no testosterone treatment (FD+T−, N = 10); female hyperandrogenic vitamin D-supplemented group (FD+T+, N = 11); female vitamin D-deficient group, no testosterone treatment (FD−T−, N = 11); female hyperandrogenic vitamin D-deficient group (FD−T+, N = 10); male vitamin D-supplemented group (MD+, N = 11), and male vitamin D-deficient group (MD−, N = 11). 

Rats were housed at constant room temperature (22 °C ± 1 °C) in a 12 h/12 h light–dark cycle. They were provided tap water ad libitum. Corresponding to their prescribed diet, specific rat chow was provided ad libitum (see below). No unexpected com-plication or side effect was observed during the chronic treatment period. 

### 4.3. Chronic Treatment of the Rats

#### 4.3.1. Vitamin D Deficiency and Supplementation

In the vitamin D-deficient (FD−T−, FD−T+, and MD−) groups, vitamin D deficiency was achieved by feeding the rats with Vitamin D Free Lab Rat/Mouse Chow (Ssniff Spezialdiaten GmbH, Soest, Germany) containing less than 5 IU/kg Vitamin D3 ad libitum for eight weeks [60,68]. 

The vitamin D-supplemented (FD+T−, FD+T+, and MD+) groups were fed regular chow containing 1000 IU/kg of vitamin D ad libitum for eight weeks. Furthermore, vitamin D-supplemented animals received additional oral (through a gavage cannula) chole-calciferol (Vigantol) treatment to achieve the target serum 25-hydroxy-cholecalciferol level. To achieve this, the oral administration of additional vitamin D was given as follows: 500 IU Vigantol on week 2, and 140 IU/100 g on weeks 4, 5, 6, 7, and 8. We used a once-a-week dose instead of a daily dose to reduce the stress on the animals. The average 25-hydroxy-cholecalciferol level at the end of chronic eight-week-long treatment was 19.66 ± 0.81 ng/mL) [28,55]. 

#### 4.3.2. Hyperandrogenism

Hyperandrogenism was induced by eight-week-long transdermal testosterone (Androgel) treatment in FD+T+ and FD−T+ groups. Testosterone was applied five times a week for eight weeks at a dose of 0.0333 mg × gram body weight. Regular body weight measurements were performed properly to adjust the dose. An amount of 5 g of Androgel contained 50 mg of testosterone. Steady state was reached on the second treatment day. Technique: A 3 × 3 cm skin area on the back was regularly shaved (2–3 days). The amount of gel applied was evenly distributed and the animal was kept separated until the gel dried (about 5 min) [25].

### 4.4. Myography

The animals were processed in the 8th week of treatment under pentobarbital anesthesia (Nembutal 45 mg/kg intraperitoneal injection). The anesthetized animals were perfused transcardially (using a syringe inserted into the left ventricle) with 150 mL of heparinized (10 IU/mL) Krebs solution. Then, the carotid arteries were dissected with careful micropreparation. Four ring segments were prepared (2–2 from both sides, each equally 3 mm long) and placed in a Petri dish containing Krebs solution for myographic measurement. Rings were mounted on a conventional wire myograph setup (610-M Multichamber Myograph System; Danish Myo Technology, Denmark/Ballagi LTD, Budapest, Hungary) as described earlier [25]. The remaining vascular tissue was fixed in formalin and embedded in paraffin (N = 4–8 in each group). 

Carotid artery rings were mounted on 120 µm diameter stainless steel needles of the myograph equipment under a preparation microscope. Each organ chamber was filled with 6 mL of Krebs solution. The temperature of the bath was kept at 37 °C, the pH was maintained at 7.4, and the bath fluid was bubbled with a carbogen gas containing 5% carbon dioxide and 95% oxygen. The initial tension of the vessels was set at 10 mN. Four segments of the experiment were run parallel. The agonist-dependent responses (contraction and relaxation) of blood vessels were tracked on computer-displayed curves. The isometric tension of the vascular segments was recorded and analyzed using the LabChart Evaluation Software Version 8 of the Powerlab data processing system (ADInstruments, Oxford, UK-Ballagi LTD, Budapest, Hungary). Vasoactive substances were dissolved in Krebs solution. All concentrations are given for the final concentration in the organ bath. Rings were gradually stretched in slow steps up to 10 mN. This resting basic isometric tension was reached in about 20 min and was followed by a minimum of 30 min of equilibration while tension was further adjusted if needed. The bubbling was constantly checked, the temperature was kept at 37 °C, which was monitored with a thermosensor, and the tension was constantly corrected. After the equilibration period, hyperkalemic Krebs solution (124 mM potassium, K+) was first added to the segments for 3 min to establish a reference contraction (100%). After the chemicals were added, the chambers were thoroughly rinsed several times with a vacuum aspirator, and then fresh Krebs solution, prewarmed in a thermostat, was refilled through pump tubes. After washing periods during the experiment, the baseline tone was always restored. After several washes and after the tensions were restored to baseline values, cumulative dose–response curves for Ang II (10^−9^–10^−7^ M) were plotted. After several washes, a dose–response relaxation curve with estrogen (E2 10^−8^–10^−5^) or testosterone (T 10^−9^–10^−6^) was recorded after precontraction with phenylephrine (100 nM). After each dose, we waited for the development of a stable contraction.

During the evaluation of wire myograph experiments, contraction data were normalized to KCl contraction (which was taken at 100%) and relaxation data were calculated as percent values of the precontraction level (with Phe, calculated as the precontraction level minus the baseline). The baseline tension before each dose–response curve was set to 10 mN, which was the reference level for the calculation of contraction and relaxation states. To evaluate contraction responses, we calculated between endpoints of the baseline and contraction levels (calculated as (contraction level-baseline level)/amplitude of KCl contraction). To evaluate relaxation responses, we calculated between endpoints of baseline and precontraction levels (with Phe) compared with the value of relaxation level (calculated as: (precontraction level-relaxation level)/(precontraction level-baseline level)). In all cases, a washout period followed dose–response curves, and then the baseline level was achieved.

### 4.5. Immunohistochemistry

The organs were fixed in formalin for 24 h and then dehydrated for a minimum of 24 h in an alcohol series (70%, 2 × 1 h with 95%, 2 × 1 h with 100%, then 3 × 20 min with Xylene). Then, the organs were embedded in a paraffin block for 3 × 1 h and sliced into 7-micron slices with a microtome (Thermofischer; Waltham, MA, USA). The slides were rehydrated for staining with a descending alcohol series as follows (2 × 10 min Xylene dissolved in paraffin and only the tissue remained on the slide): 100%, 100%, 96%, 70%, followed by 5 min in distilled water. Staining densities of Angiotensin II type I receptor (ATR1), androgen receptor (AR), estrogen-alpha receptor (ERα), and vitamin D receptor (VDR) were investigated using immunohistochemistry. 

Antigen retrieval was achieved by boiling in citrate buffer (pH = 6 for 15 min, followed by cooling for 40 min), followed by washing in phosphate-buffered saline (PBS). The wash took 3–5 min in each case.

For a period of 10 min, H_2_O_2_ (3%) was used to neutralize endogenous peroxidase activity (to avoid cross-reaction at the seconder). Non-specific labeling was prevented by using 2.5% normal horse serum blocking (Vector Biolabs, Burlingame, CA, USA) for one hour. The following primary antibodies were administered: Anti ERα receptor rabbit polyclonal 1:85 (Millipore; Burlington, MA, USA), Anti-AT1 receptor rabbit monoclonal 1:200 (Bioss Antibodies; Woburn, MA, USA), Anti-Androgen receptor (AR) rat monoclonal antibody 1:100 (Invitrogen; Waltham MA, USA), and Vitamin D receptor (VDR) mouse monoclonal antibody 1:200 (Santa-Cruz biotechnology; Dallas, TX, USA). The primary antibody was dissolved in a blocking solution (factory or Triton X PBS). A minimum of 8 h (overnight) of incubation at 4 °C was performed, with the exception of AR, where this was performed for 3 h at 37 °C. This was followed by washing with Triton X PBS for 3 × 5 min in the morning. The selection of the vector seconder antibody depends on the animal in which the primer is produced: secondary labeling was performed via horseradish peroxidase (HRP)-linked anti-rabbit (ERα, ATR1), anti-mouse (VDR) horse antibodies (Vector Biolabs, Burlingame, CA, USA), or anti-rat (AR) goat antibody (Vector Biolabs, Burlingame, CA, USA). 

A 40 min incubation with 1 × 5 min TX PBS was followed by 2 × 5 min of plain PBS washout. Visualization was achieved by brown-colored 3,3′-diaminobenzidine (DAB) (Vector Biolabs, Burlingame, CA, USA). As DAB bound to the secondary antibody, brown precipitation developed in 2–10 min. The DAB is only stopped by distilled water, so this was used for a 3 × 5 min wash. Purple-colored hematoxylin (hematoxylin modified to Gill II, Sigma-Aldrich, St. Louis, MO, USA) was applied for counterstaining. This was stopped by washing with alkaline tap water.

After washing with distilled water, the samples were treated with a series of rising alcohols (70%, 90%, 100%, and 100% alcohol for 3-3-3-3 min, followed by xylene for 5 min), and then finally sealed with DPX cover.

Slides were photographed with a Nikon eclipse Ni-U microscope equipped with a DS-Ri2 camera (Nikon, Minato-Tokyo Japan). Photos of the ER-α slides were taken in 20× magnification; in other cases (Vitamin D, ATR1, Androgen) 10× magnification was set.

Image J (FIJI^®^) software (https://imagej.net/software/fiji/downloads, accessed on 18 November 2023, National Institutes of Health, Bethesda, MA, USA). was used to evaluate the results of the immunohistochemistry analysis by separating the background staining (DAB + hematoxylin). We determined the non-calibrated optical density (OD) of the DAB.

### 4.6. Statistics

The results are presented as mean ± (SEM) (standard error of the mean). *p* < 0.05 probability was considered a statistically significant difference. Normal distribution was tested using the Shapiro–Wilk test.

For the analysis of vascular function curves, the following methods were used: (1) repeated measured two-way ANOVA with Bonferroni’s post hoc test (compared with the MD+ and MD− groups); (2) two-way ANOVA at the same concentration was completed with Tukey’s post hoc test (factor1: testosterone treatment; factor2: vitamin D status; compared with the FD+T−, FD+T+, FD−T− and FD−T+ groups); and (3) two-way ANOVA at the same concentration was completed with Tukey’s post hoc test (factor1: vitamin D status; factor2: gender; compared with the male and female groups) using GraphPad Prism 8 (GraphPad Software PRISM 9.5.0., San Diego, CA, USA).

For quantitative evaluation of immunohistochemical sections, statistical comparisons were made using the following methods: (1) two-way ANOVA with Tukey’s post hoc test (factor1: testosterone treatment; factor2: vitamin D status; compared with the FD+T−, FD+T+, FD−T− and FD−T+ groups); (2) two-way ANOVA with Tukey’s post hoc test (factor1: vitamin D status; factor2: gender; compared with the male and female groups) with Tukey’s post hoc test, (3) Kruskal–Wallis test with Dunn’s multiple-comparison test (in the case of fewer than three cases); and (4) unpaired t test (compared with the MD+ and MD− groups).

## 5. Conclusions

This is the first study of the combined effects of testosterone and vitamin D in females and males. Steroids affect each other’s receptor expression and vascular reactivity which can differ also upon gender. This is a new concept in studying the effects of testosterone and vitamin D in females and males.

Chronic testosterone treatment significantly enhanced angiotensin II-induced contraction in our PCOS model. Vitamin D deficiency decreased the estrogen-induced relaxation in female animals. The testosterone-induced relaxation also decreased in vitamin D-deficient females who did not receive chronic testosterone treatment, but FD−T+ animals avoided this decline. The density of estrogen receptors rises in response to vitamin D insufficiency, whereas the density of androgen receptors rises in response to long-term testosterone therapy. The effects of estrogen and testosterone on relaxation also diminished in males with vitamin D deficiency. As a sex difference, the angiotensin II-induced contraction is considerably higher in males than in the female groups. Females have a substantially larger density of estrogen receptors.

In our PCOS model, the vascular reactivity of female animals was midway between vitamin D-treated males and control females during angiotensin II-induced contraction. Estrogen-induced relaxation and testosterone-induced relaxation showed gender differences in FD−T− females compared with the MD+ group. In contrast, no gender difference was observed in testosterone-treated females compared with males. This may be due to increased androgen receptor density caused by chronic androgen treatment.

Vitamin D deficiency impairs relaxation in males and non-hyperandrogenic females. Chronic androgen treatment increases the vasoconstrictor response of the carotid artery. Vitamin D deficiency and chronic androgen treatment show significant gender differences in carotid artery reactivity.

According to our findings, both vitamin D deficiency and PCOS (chronic testosterone effect) result in a dysfunctional vascular response, which can contribute to cardiovascular diseases. Since PCOS is often associated with vitamin D deficiency, it is very important to screen this patient population for vitamin D levels and, in case of deficiency, to supplement early and maintain vitamin D levels at optimal levels. This will help reduce cardiovascular risk in women with PCOS.

## Figures and Tables

**Figure 1 ijms-24-16577-f001:**
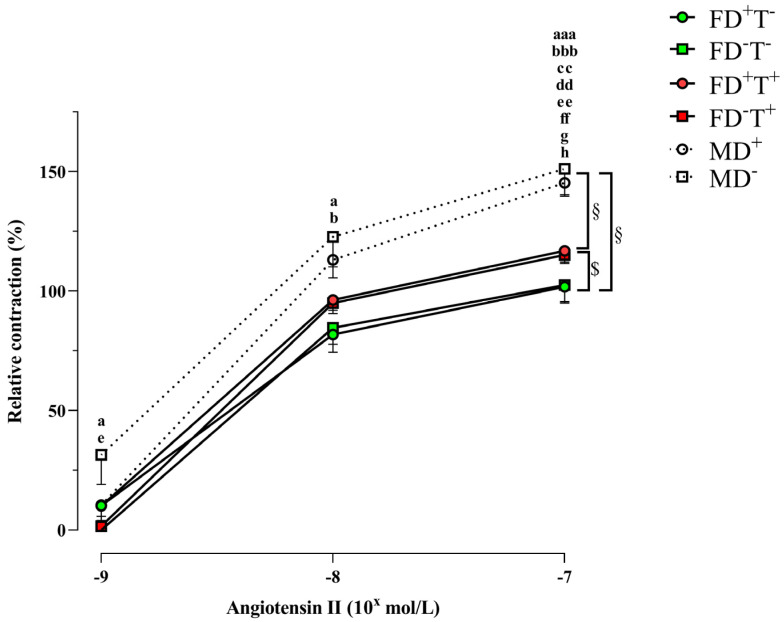
Angiotensin II-induced contractility of the isolated vascular segments from the carotid artery from each group. Females: Chronic testosterone treatment led to a more pronounced contraction in the female groups independently of vitamin D status: two-way ANOVA at the same concentrations (factor1: vitamin D status (not significant), factor2: chronic testosterone treatment ($, *p* < 0.05)) with Tukey’s post hoc test. Males: In male groups, the angiotensin II-induced contraction also did not differ between vitamin D-deficient and supplemented groups: two-way repeated measures ANOVA with Bonferroni’s post hoc test. Male/female comparison: Male sex was associated with more pronounced relative contraction than females both in testosterone-untreated and testosterone-treated groups independently of vitamin D status. Androgel untreated: two-way ANOVA at the same concentrations (factor1: vitamin D status (not significant), factor2: sex (§, *p* < 0.05), effect size f = 0.553, effect size d = 1.485 at 10^−7^ concentrations) with Tukey’s post hoc test: a, *p* < 0.05 MD− vs. FD−T−; aaa, *p* < 0.001 MD− vs. FD−T−; b, *p* < 0.05 MD− vs. FD+T−; bbb, *p* < 0.001 MD− vs. FD+T−; cc, *p* < 0.01 MD+ vs. FD+T−; dd, *p* < 0.01 MD+ vs. FD−T−. Androgel treated: (factor1: vitamin D status (not significant), factor2: sex (§, *p* < 0.05), effect size f = 0.713, effect size d = 1.914 at 10^−7^ concentrations) with Tukey’s post hoc test: e, *p* < 0.05 MD− vs. FD−T+; ee, *p* < 0.01 MD− vs. FD−T+; ff, *p* < 0.01 MD− vs. FD+T+; g, *p* < 0.05 MD+ vs. FD+T+; h, *p* < 0.05 MD+ vs. FD−T+. Results are presented as mean ± SEM; n = 9–11 for each group. Abbreviations: FD+T−, female vitamin D-supplemented group, no testosterone treatment; FD−T−, female vitamin D-deficient group, no testosterone treatment; FD+T+, female hyperandrogenic and vitamin D-supplemented group; FD−T+, female hyperandrogenic and vitamin-D-deficient group; MD^+^, male vitamin D-supplemented group; MD−, male vitamin D-deficient group.

**Figure 2 ijms-24-16577-f002:**
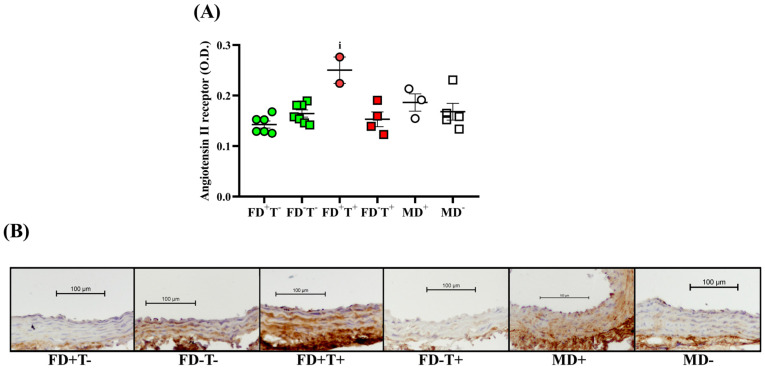
Angiotensin II receptor immunohistochemistry. (**A**) Angiotensin receptor optical density (OD). In female groups, the Kruskal–Wallis test with Dunn’s post hoc test was used: i, *p* < 0.05, FD+T− vs. FD+T+. The ATR1 OD did not differ between MD+ and MD− groups, unpaired *t* test. There were no sex differences between male and female groups (two-way ANOVA with Tukey’s post hoc test (FD+T−, FD−T− vs. MD+, MD−); Kruskal–Wallis test with Dunn’s post hoc test (FD+T+, FD−T+ vs. MD+, MD−). The results are presented as individual data points; horizontal lines represent mean ± SEM. n = 2–7 in each group. (**B**) Images representative of angiotensin II receptor-stained carotid artery segments from the FD+T−, FD−T−, FD+T+, FD−T+, MD+, and MD− animal groups. Scale bar, 100 µm. Abbreviations: FD+T−, female vitamin D-supplemented group, no testosterone treatment; FD−T−, female vitamin D-deficient group, no testosterone treatment; FD+T+, female hyperandrogenic and vitamin D-supplemented group; FD−T+, female hyperandrogenic and vitamin D deficient group; MD+, male vitamin D supplemented group; MD−, male vitamin D deficient group.

**Figure 3 ijms-24-16577-f003:**
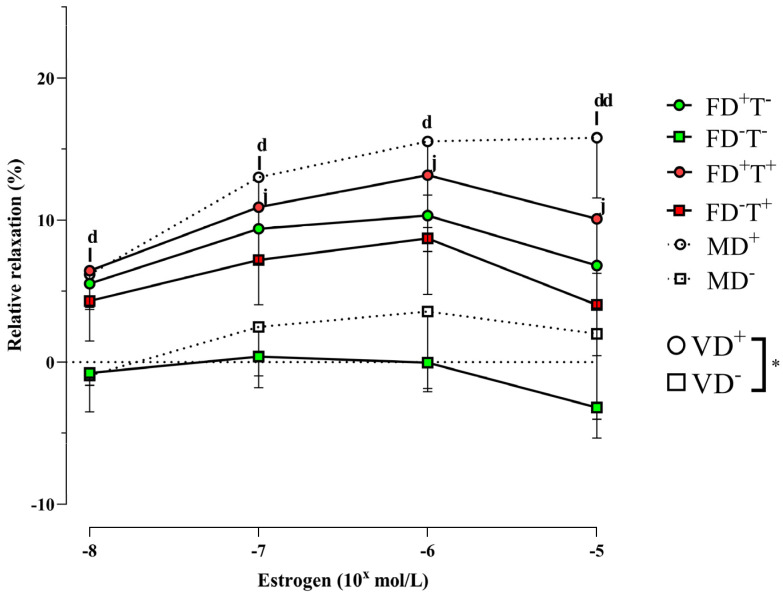
Vasodilator capacity of estrogen on isolated carotid artery segments in each group. Females, vitamin D deficiency: Vitamin D deficiency was associated with reduced relaxation in female groups regardless of chronic testosterone treatment: two-way ANOVA at the same concentrations (factor1: vitamin D status (*, *p* < 0.05), factor2: chronic testosterone treatment (not significant)), with Tukey’s post hoc test: j, *p* < 0.05 FD−T− vs. FD+T+. Males, vitamin D deficiency: vitamin D deficiency was associated with reduced estradiol relaxation in males (*, *p* < 0.05 VD+ vs. D−): two-way repeated measures ANOVA with Bonferroni’s post hoc test. Males compared with the female groups: Two-way ANOVA at the same concentrations (factor1: vitamin D status (*, *p* < 0.05), factor2: sex), with Tukey’s post hoc test, not different. Vitamin D sensitivity is not dependent on sex, only FD−T− rats show smaller estrogen-induced relaxation than MD+ animals. Tukey’s post hoc test: d, *p* < 0.05 MD+ vs. FD−T−; dd, *p* < 0.01 MD+ vs. FD−T−; l, *p* < 0.05 MD+ vs. MD−. Results are presented as mean ± SEM; n = 9–11 for each group. Abbreviations: FD+T−, female vitamin D-supplemented group, no testosterone treatment; FD−T−, female vitamin D-deficient group, no testosterone treatment; FD+T+, female hyperandrogenic and vitamin D-supplemented group; FD−T+, female hyperandrogenic and vitamin D-deficient group; MD+, male vitamin D-supplemented group; MD−, male vitamin D-deficient group.

**Figure 4 ijms-24-16577-f004:**
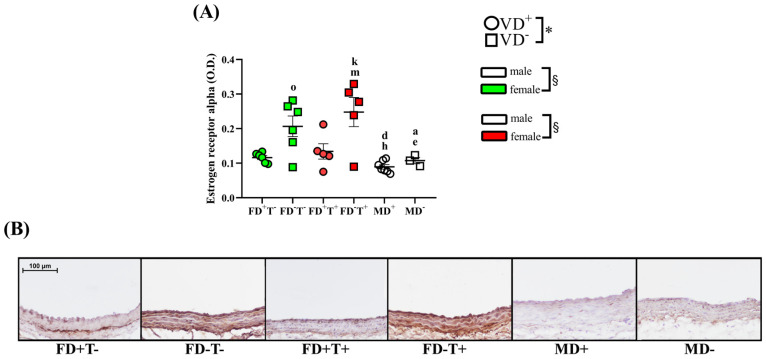
Estrogen receptor immunohistochemistry. (**A**) Estrogen receptor OD. Females, vitamin D deficiency: Vitamin D deficiency increased the estrogen receptor optical density in female groups independently of Androgel treatment: two-way ANOVA (factor1: vitamin D status (*, *p* < 0.05), factor2: chronic testosterone treatment (not significant)) with Tukey’s post hoc test: k, *p* < 0.05 FD+T+ vs. FD−T+; m, *p* < 0.05 FD+T− vs. FD−T+: o, *p* < 0.05 FD+T− vs. FD−T−. Males, vitamin D deficiency: In males, estrogen receptor OD did not differ between vitamin D treatment groups: unpaired t test. Female/male comparison, vitamin D deficiency: When males were compared with testosterone-untreated females, female sex and vitamin D deficiency were associated and significantly increases in the estrogen receptor OD: two-way ANOVA (factor1: vitamin D status (*, *p* < 0.05), factor2: sex (§, *p* < 0.5)), with Tukey’s post hoc test, a, *p* < 0.05 MD− vs. FD−T−; d: *p* < 0.05 MD+ vs. FD−T−. This was not altered by testosterone treatment of the females. Two-way ANOVA (factor1: vitamin D status *p* < 0.05, factor2: sex (§, *p* < 0.05)) with Tukey’s post hoc test: e, *p* < 0.05 MD− vs. FD−T+; h, *p* < 0.05 MD+ vs. FD−T+. Data are shown as individual data points; horizontal lines represent mean ± SEM. n = 3–7 in each group. (**B**) Representative images of estrogen receptor alpha-stained aorta from FD+T−, FD−T−, FD+T+, FD−T+, MD+, and MD− groups, respectively. Scale bar, 100 µm. Abbreviations: FD+T−, female vitamin D-supplemented group, no testosterone treatment; FD−T−, female vitamin D-deficient group, no testosterone treatment; FD+T+, female hyperandrogenic and vitamin D-supplemented group; FD−T+, female hyperandrogenic and vitamin D-deficient group; MD+, male vitamin D-supplemented group; MD−, male vitamin D-deficient group.

**Figure 5 ijms-24-16577-f005:**
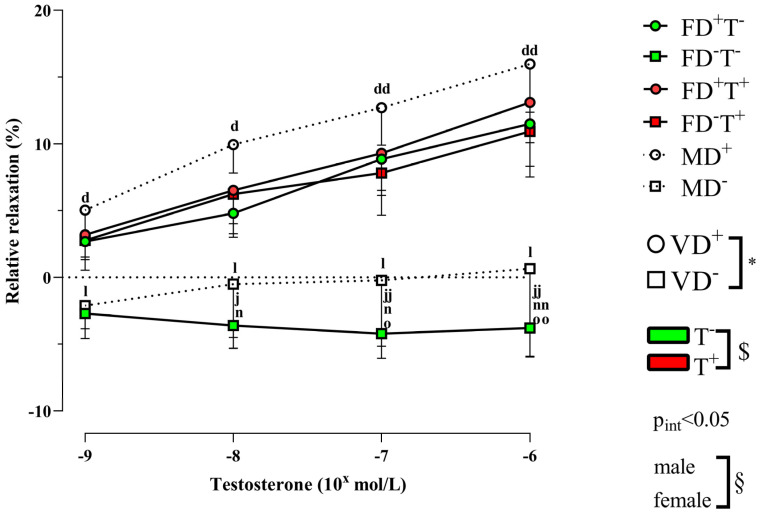
Vasodilator capacity of testosterone on isolated carotid artery segments in each group. Females, vitamin D deficiency, testosterone dependency: In female groups, vitamin D deficiency reduced testosterone relaxation but only in animals not receiving Andogel treatment. There is an interaction between the two factors (two-way ANOVA, at the same concentration, factor1: vitamin D status (*, *p* < 0.05), factor2: Androgel treatment ($, *p* < 0.05), interaction: (p_intection_, *p* < 0.05)) with Tukey’s post hoc test: j, *p* < 0.05 FD+T+ vs. FD−T−; jj, *p* < 0.01 FD+T+ vs. FD−T−; n, *p* < 0.05 FD−T− vs. FD−T+; nn, *p* < 0.01 FD−T− vs. FD−T+; o, *p* < 0.05 FD+T− vs. FD−T−; oo, *p* < 0.01 FD+T− vs. FD−T−, FD+T+ vs. FD−T+, n.s. Males, vitamin D deficiency. In males, vitamin D deficiency was associated with reduced testosterone relaxation: two-way repeated measures ANOVA with Bonferroni’s post hoc test. l, *p* < 0.05 MD+ vs. MD−. Male/testosterone untreated female comparison. Male compared with the female Androgel untreated groups, vitamin D deficiency associated with smaller testosterone-induced relaxation: two-way ANOVA at the same concentrations (factor1: vitamin D status (*, *p* < 0.05), factor2: sex (§, *p* < 0.05), effect size f = 0.482, effect size d = 1.294 at 10^−6^ concentration), Tukey’s post hoc test: d, *p* < 0.05 MD+ vs. FD−T−; dd, *p* < 0.01 MD+ vs. FD−T−; l, *p* < 0.05 MD+ vs. MD−. Male/testosterone-treated female comparison. Male compared with the Androgel-treated female groups, vitamin D deficiency associated with smaller testosterone-induced relaxation: two-way ANOVA at the same concentrations (factor1: vitamin D status (*, *p* < 0.05), factor2: sex (§, *p* < 0.05), effect size f = 0.409, effect size d = 1.096 at 10^−6^ concentrations)) with Tukey’s post hoc test: l, *p* < 0.05 MD+ vs. MD−. Results are presented as mean ± SEM; n = 9–11 for each group. Abbreviations: FD+T−, female vitamin D-supplemented group, no testosterone treatment; FD−T−, female vitamin D-deficient group, no testosterone treatment; FD+T+, female hyperandrogenic and vitamin D-supplemented group; FD−T+, female hyperandrogenic and vitamin D-deficient group; MD+, male vitamin D-supplemented group; MD−, male vitamin D-deficient group.

**Figure 6 ijms-24-16577-f006:**
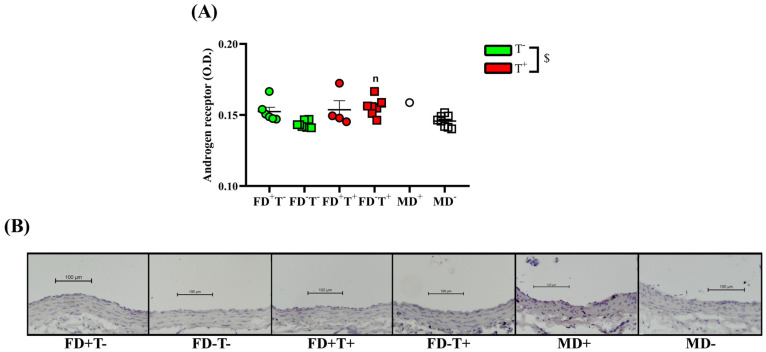
Androgen receptor immunohistochemistry. (**A**) Androgen receptor OD. Females, testosterone effect. In female groups, Androgel treatment was associated with increased androgen receptor optical density: two-way ANOVA (factor1: vitamin D saturation status (n.s.), factor2: Androgel treatment ($, *p* < 0.05)) with Tukey’s post hoc test: n, *p* < 0.05 FD−T− vs. FD−T+. Data are presented as individual data points; horizontal lines represent mean ± SEM. n = 1–7 in each group. (**B**) Images representative of the androgen receptor-stained aorta from FD+T−, FD−T−, FD+T+, FD−T+, MD+, and MD− groups. Scale bar, 100 µm. Abbreviations: FD+T−, female vitamin D-supplemented group, no testosterone treatment; FD−T−, female vitamin D-deficient group, no testosterone treatment; FD+T+, female hyperandrogenic and vitamin D-supplemented group; FD−T+, female hyperandrogenic and vitamin D-deficient group; MD+, male vitamin D-supplemented group; MD−, male vitamin D-deficient group.

## Data Availability

The published article contains all generated and analyzed data from this series.

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
