# Peer review of "Effects of Gender and Vitamin D on Vascular Reactivity of the Carotid Artery on a Testosterone-Induced PCOS Model"

_ijms, 2023, doi:10.3390/ijms242316577_

Round 1

Reviewer 1 Report

Comments and Suggestions for Authors

ID: ijms-2703516

Effects of gender and vitamin D on vascular reactivity of carotid artery on a testosterone induced PCOS model. by Süli A et al.

To the Authors:

General comments:

The authors investigated the effect of polycystic ovary syndrome and vitamin D deficiency (VDD) in carotid artery of male and female Wistar rats.  The results suggested the combined effects of testosterone and vitamin D, in which steroids affect each other’s receptor expression and vascular reactivity which can differ also upon gender.  It was considered that the study was well structured, and the result included novelty; however, several points should be addressed to improve the manuscript.

Specific comments:

1. The authors utilized a model of PCOS by administration of androgen; however, the pathophysiology of PCOS involves insulin tolerance, diabetes, obesity, and hypothalamo-pituitary dysfunction, and so on.  Please address the rationale regarding the present model induced by androgen as a PCOS model in rats.

2. The authors should quantify the mRNA/protein levels of the angiotensin II receptor (Fig. 2), estrogen receptor (Fig. 4), and androgen receptor (Fig. 5) in addition to the immunohistochemical assessments.

3. Did the authors measure the serum level of calcium and 1,25-dihydroxyvitamin D in a VDD model?  Please discuss the molecular mechanisms of vitamin D metabolism and the results of myograph measurements shown in Fig. 1, 3, and 5.

4. How were the changes of expression levels of androgen receptors and VD receptors in the process of contractile changes on the vascular reactivity?

5. Please add the figure that summarizes the results obtained in the present study.

Reviewer 2 Report

Comments and Suggestions for Authors
  1. Context and Rationale:
    • The abstract sets the stage by identifying PCOS and VDD as risk factors for cardiovascular disease, which is a strong start. However, the rationale for using male and female rats in a PCOS study must be explained. Typically, PCOS is a female-specific syndrome, so the inclusion of male subjects needs justification. Does the study aim to understand the broader hormonal influences on the cardiovascular system?
  2. Objective:
    • The study aims to investigate the effects of PCOS and VDD on the carotid artery in rats. This objective is clearly stated but could be enhanced by indicating the relevance of these effects to the broader understanding of cardiovascular risks associated with PCOS and VDD.
  3. Methodology:
    • The methods section mentions the induction of PCOS and VDD, myograph measurements and immunohistochemistry. However, some missing elements could be addressed:
      • Induction of PCOS: How does the Androgel dosage translate to human PCOS conditions?
      • Vitamin D Status: What are the specific criteria for VDD and the parameters for vitamin D supplementation? How do these relate to human conditions?
      • Analytical Techniques: More detail on the myograph measurements and immunohistochemical analyses could be helpful. What were the specific endpoints measured?
  4. Results:
    • The results suggest significant gender differences in the vascular response to hormones and vitamin D status. This is a crucial finding but needs statistical significance and effect size clarification.
    • The statement about "significantly enhanced angiotensin-induced contraction" could be improved by including information on the magnitude of this effect.
    • Clarification is needed on whether the decreased estrogen-induced relaxation in the context of VDD is a relative change or absolute and how this compares to normal levels.
  5. Interpretation:
    • The interpretations made from the immunohistochemistry findings (e.g., changes in AR and ERα density) are interesting but require context. How do these changes compare with a healthy baseline?
    • There is a mention of gender differences without a clear explanation of what these differences entail in terms of vascular health or PCOS pathology.
  6. Conclusion:
    • It concludes that VDD and PCOS lead to vascular dysfunction, potentially contributing to cardiovascular diseases. This conclusion is logical based on the results provided but should briefly address the implications of these findings for treatment or prevention strategies in the human population.

General Suggestions:

  • The abstract would benefit from a more detailed explanation of the significance of the study's findings for the current understanding of PCOS and cardiovascular risk.
  • It may be useful to compare and contrast the findings with existing literature to highlight the study's contribution to the field.
  • A brief discussion of limitations, such as using an animal model and its implications for human health, would be valuable.

Comments on the Quality of English Language

Minor editing

Round 2

Reviewer 1 Report

Comments and Suggestions for Authors

ID: ijms-2703516

Effects of gender and vitamin D on vascular reactivity of carotid artery on a testosterone induced PCOS model. by Süli A et al.

To the Authors:

General comments:

The authors revised the manuscript appropriately according to the suggestions.

Reviewer 2 Report

Comments and Suggestions for Authors

I want to congratulate the authors for their efforts in revising this manuscript. No more comments.